# CoVM^2^: Molecular Biological Data Integration of SARS-CoV-2 Proteins in a Macro-to-Micro Method

**DOI:** 10.3390/biom12081067

**Published:** 2022-08-02

**Authors:** Hongjun Chen, Xiaotian Hu, Yanshi Hu, Jiawen Zhou, Ming Chen

**Affiliations:** 1Department of Bioinformatics, College of Life Sciences, Zhejiang University, Hangzhou 310058, China; chenhj@zju.edu.cn (H.C.); wuuwst@zju.edu.cn (X.H.); yanshihu@zju.edu.cn (Y.H.); 2Chu Kochen Honors College, Zhejiang University, Hangzhou 310058, China; 3200103629@zju.edu.cn; 3Institute of Hematology, Zhejiang University School of Medicine, The First Affiliated Hospital, Zhejiang University, Hangzhou 310058, China

**Keywords:** SARS-CoV-2, virus–host interactions, protein structure docking, binding affinity, molecular biology, data integration

## Abstract

The COVID-19 pandemic has been a major public health event since 2020. Multiple variant strains of SARS-CoV-2, the causative agent of COVID-19, were detected based on the mutation sites in their sequences. These sequence mutations may lead to changes in the protein structures and affect the binding states of SARS-CoV-2 and human proteins. Experimental research on SARS-CoV-2 has accumulated a large amount of structural data and protein-protein interactions (PPIs), but the studies on the SARS-CoV-2–human PPI networks lack integration of physical associations with possible protein docking information. In addition, the docking structures of variant viral proteins with human receptor proteins are still insufficient. This study constructed SARS-CoV-2–human protein–protein interaction network with data integration methods. Crystal structures were collected to map the interaction pairs. The pairs of direct interactions and physical associations were selected and analyzed for variant docking calculations. The study examined the structures of spike (S) glycoprotein of variants Delta B.1.617.2, Omicron BA.1, and Omicron BA.2. The calculated docking structures of S proteins and potential human receptors were obtained. The study integrated binary protein interactions with 3D docking structures to fulfill an extended view of SARS-CoV-2 proteins from a macro- to micro-scale.

## 1. Introduction

COVID-19 has sparked a global pandemic since the year 2020. A coronavirus was identified as the causative agent of COVID-19 and named SARS-CoV-2 [1]. Systematic studies including sequencing, protein interactions, and structural docking have been carried out to identify SARS-CoV-2-related proteins. The related virus–human protein interactions occurred in the progression of a SARS-CoV-2 infection [2]. The spike glycoprotein became the focus of research due to its mediating role in virus entry into cells by binding with human ACE2 receptors [3]. When SARS-CoV-2 infects the host cell, the S protein binds to the host receptor ACE2, triggering the virus-cell fusion process [4].

The sequence of SARS-CoV-2 has undergone frequent mutations, and multiple mutant strains have been distinguished based on a large number of mutational characteristics [5]. The mutation site of variant strains Delta B.1.617.2, Omicron BA.1, and Omicron BA.2 have been investigated by multiple studies and recorded in detail in public databases [6], thus, the sequences of the variants can be collected. In addition to ACE2, several human proteins with potential S-protein receptor function have also been experimentally identified [7,8]. Tyrosine-protein kinase receptor UFO (AXL), Neuropilin-1 (NRP1), and CD209 act as SARS-CoV-2 receptors which play critical roles in SARS-CoV-2 invasion and infection [9,10,11,12].

The protein–protein interactions (PPIs) between SARS-CoV-2 and human receptor candidates have been detected by experimental methods. Affinity purification-mass spectrometry (AP-MS) [13] has been a widely-used method to identify PPIs and protein complexes by generating bait-prey data sets [14] and is suitable for detecting virus–host multiprotein complexes [15]. Proximity-dependent biotin identification (BioID-MS) [16] is a method designed for the detection of transient PPI that frequently emerged during the viral infection. Gordon et al. [17] obtained 332 pairs of high-confidence virus–human protein interactions using AP-MS and identified 66 human proteins that could be used for drug therapy. Using AP-MS and BioID-MS experimental methods, Liu et al. [15] mapped the interaction between viral proteins and host proteins, resulting in 693 hub proteins that can be used for drug repurposing. Public databases including IntAct [18], VirHost [19], and RCSB PDB [20] have collected the annotation information and structural data of related proteins. There have been repositories of comprehensive SARS-CoV-2-related information. Ahsan et al. proposed OverCOVID [21], an integrative server including SARS-CoV-2-related databases, web servers, and tools. Satyam et al. established COVIDium [22], which provided similar information, but also added PPI network data sets and classified all the data entries.

The docking structures analyzed in previous studies are based on the sequence of the wildtype (WT) strain of SARS-CoV-2. Wierbowski et al. [23] constructed a 3D-structured SARS-CoV-2 interactome showing the calculated binding interface of multiple proteins within the interaction between viral docking and human interactors. Barh et al. [24] detected gradually increasing transmissibility and declining pathogenicity of SARS-CoV-2 WT and variants Gamma, Delta, and BA.1 through sequence and structure analysis. The docking of BA.2 variant proteins with host receptor proteins has not been systematically analyzed and obtaining the experimental docking structures requires extensive experiments. Furthermore, these datasets require further integration and adoption of the novel mutant sequences of SARS-CoV-2 variants.

In this study, an interaction network with 2385 pairs of interactions between 29 viral proteins and 1144 human proteins was constructed. Among the 196 interaction pairs of the S protein, the crystal structures and protein complexes were collected. On the other hand, for the direct interactions and physical associations that may have structural binding in the integrated network, the structures of the two interacting proteins were collected respectively for docking. In comparison with the crystal structures and protein complexes of the SARS-CoV-2 wildtype (WT) strain, this study obtained some reliable computational docking structures of the S protein of the variant strain and the human receptor protein. The proposed integrated framework of the PPI network (macro view) and the virus–host protein binding (micro view) was named CoVM^2^. Calculations of affinities for these structures revealed differences from the WT strain proteins. Figure 1 illustrates the workflow of the research.

## 2. Materials and Methods

### 2.1. Collection of Datasets and Network Construction

The SARS-CoV-2 and human interaction pairs were obtained from Liu’s work [15], IntAct [18], and VirHost [19]. All the interaction data from different sources were integrated to construct the protein interaction network. The PPI network was integrated and visualized by Cytoscape [25]. Node degree is one of the network’s topological features indicating how many interactions are connected with a protein in the PPI network to identify specific target proteins [26]. The strength of interaction is determined through the detection methods, which can be a measurement of an exact PPI or can imply the affinity of the proteins [27]. Here, the degree was regarded as a necessary parameter to screen the human proteins that are more likely to interact with an individual SARS-CoV-2 protein. Metascape [28] analysis was conducted on the subgroup of S protein’s isolated interactions (node degree is 1) for validation in biological processes.

All protein sequences of SARS-CoV-2 and human are retrieved from UniProt [6]. The canonical sequence of SARS-CoV-2 WT S protein was retrieved from UniProt ID P0DTC2. It was then processed by an in-house python script to replace the amino acids in the mutation site to generate the variant sequences. The 3D structures of full-length proteins or partial regions were retrieved in the Protein Data Bank (PDB) database [20].

### 2.2. Homology Modeling for SARS-CoV-2 Variants

Other than the main viral proteins with 3D structures that have been solved experimentally, some of the variant strains’ structures have not been solved at the time this study was completed. Modeling of the structurally unknown viral proteins was performed with SWISS-MODEL [29]. Homology templates were sorted with multiple evaluation metrics such as global model quality estimation (GMQE) and sequence identity. The S protein receptor-binding domain (RBD) is composed of 223 residues from 319 to 541 in its sequence. The RBD of Omicron BA.2 was modeled by the template of 7WBP, chain B [30]. Chain B (319–541) of the structures 6M0J, 7WBQ, and 7WBP were extracted in PyMOL as the S protein RBD of WT, Delta, and Omicron BA.1, respectively [30,31].

### 2.3. Quality Assessment of Protein Structures

Procheck in the Structural Analysis and Verification Server (SAVES) [32] was applied to verify the modeled structure. Previous experiments suggested that structures with over 90% residues in the most favored regions of the Ramachandran plot can be regarded as accurate models [33,34]. However, the cutoff set in our research influenced the availability of models of the required PPIs. There were not sufficient native structures meeting the threshold either based on the resolution. Hence, in this research structures with less than 80% of residues in the most favored regions were filtered out according to Williams et al. [35]. In addition, ProSA-web-Protein Structure Analysis [36] is leveraged to verify whether the predicted structure has a required z-score. Usually the z-score is below zero, and it should be within the range of scores typically found in similar-size proteins solved by X-rays or NMR [35]. The root mean square deviation value (RMSD) is used to measure the structural similarity of the backbone compared with the experimentally solved template structure. Structures with a lower RMSD value are considered reliable [37], and values less than 2 Å would be considered similar. Due to lack of available crystal structures as models, protein structures that satisfy any of the quality assessments, which reach the best of a single parameter will be preserved for downstream docking calculation. We chose to keep the models which at least satisfied the z-score calculation.

### 2.4. Molecular Docking of SARS-CoV-2–Human Protein Pairs

The docking of the SARS-CoV-2–human protein pairs was performed by the HDOCK standalone package [38]. For each docking pair, the human protein was set as the receptor, and the viral protein was the ligand to generate docking structure models. The models with the lowest docking scores were screened as the candidate docking structures. The candidates were selected by comparison with experimental structures by the RMSD value by the align function in PyMOL. The PRODIGY program [39] was used to measure the binding affinity of different ligands and receptors.

## 3. Results

### 3.1. SARS-CoV-2–Human Protein Interaction Network

The integrated SARS-CoV-2–human protein interaction network was constructed and visualized (Figure 2A). Physical associations, direct interactions, and interactions of SARS-CoV-2 and 693 hub human proteins from Liu’s research [15] were integrated. Finally, 2385 pairs of interactions between 29 SARS-CoV-2 proteins and 1144 human proteins were included in the network. Annotations of nodes (proteins), edges (PPIs), and topology features were achieved from the NetworkAnalyzer and UniProt (Appendix A).

### 3.2. Biological Process Enrichment of Spike Glycoprotein Subnetwork

S protein and 196 interacting human proteins were extracted from the whole PPI network (Figure 2B). The isolated interactions of S protein included 46 human proteins with no interactions with other viral proteins, which were under gene ontology enrichment analysis. By biological process, AXL, EGFR, LDLR, NRP1, CLEC4M, WWP1, WWP2, HAVCR1, CD209, and ACE2 were enriched in the term “viral entry into host cell” (GO:0046718), indicating the enriched cluster’s potential role in SARS-CoV-2 invasion into cells (Figure 2B, Appendix A). Evidence of these proteins functioning as coronavirus spike receptors has been collected. CD209 is known as the receptor of severe acute respiratory syndrome (SARS) spike protein [40]. CLC4M is another receptor of SARS spike protein [41]. The structures of potential SARS-CoV-2-invasion-related proteins were collected as receptors for the docking calculation (Table 1).

### 3.3. Crystal Structures of Complexes of Binding Pairs

Forty-two binding structures of PPI pairs from multiple studies [10,30,31,42,43,44,45,46,47,48,49,50,51] were retrieved from the RCSB PDB database (Appendix A). Four structures with non-variant SARS-CoV-2 proteins were selected for mapping with the edges in the PPI network as the demonstration of the coronavirus macro-to-micro (CoVM^2^) view (Figure 3), and the 3D structures were presented by Mol* viewer [52]. Structures of the variant spike glycoproteins bound with ACE2 protein were also collected. The S protein binding structures of the variants Delta B.1.617.2 and Omicron BA.1 (B.1.1.529) are available as 7WBP and 7WBQ from Han’s research [30] where the structure of Omicron BA.2 was not included. Quality assessment was run on the models of SARS-CoV-2 S proteins and the interactors (Table 2). The sequence identity of SARS-CoV-2 Omicron BA.2 S protein with the protein sequence in model 7WBP is 97.76%. The binding affinity values of different variant S proteins with ACE2 were calculated, and binding interfaces were highlighted in Figure 4.

### 3.4. The Predicted Docking Structures of Potential S-Protein-Binding Pairs

The variant S proteins’ structures were used as ligands to perform the docking processes with the human receptors. Thirty-six docking structures of nine human proteins with S proteins of four SARS-CoV-2 variants were modeled. Each pair of SARS-CoV-2–human docking structures generated 100 models via the HDOCK program. Among the models, the model with the lowest score was selected. The structure of the Omicron BA.2 variant S protein binding with ACE2 had an RMSD value of 0.43 with the experimental structure. The free energy values of the selected docking structures were calculated by PRODIGY (Appendix A). For similar formations in the same viral–human docking pairs, the binding affinities varied among different variants (Figure 4E). The binding states were assigned to the PPI network for revealing the patterns by different variant subnetworks. These structures can serve as reference data for the docking of SARS-CoV-2–human proteins that have not been solved by experimental methods.

### 3.5. The Macro-to-Micro View of the SARS-CoV-2–Human Protein Associations

The coronavirus macro-to-micro (CoVM^2^) view is a PPI network integration with possible docking structures of SARS-CoV-2–human PPI pairs. Experimental and computational docking structures have been mapped to corresponding interactions. The alterations of the SARS-CoV-2 variant S protein interaction subnetwork under the differences in their binding states were achieved. OverCOVID [21] is a web portal for integrating and sharing bioinformatics resources and information on COVID-19. It has included publicly available webservers, databases, and tools associated with SARS-CoV-2 on its resource page. The CoVM^2^ network and structural datasets are available to access on OverCOVID’s new page (http://bis.zju.edu.cn/overcovid/covm2, accessed on 7 June 2022), which could be a great expansion of this field of research. At the top of the website is the visualization of the integrated SARS-CoV-2–human PPI network. By clicking the panel, users can view the interactive functions such as the protein detail tables. Users can scroll down the page to find the download link of the data sets including network, structural details of virus–human protein binding, and python scripts.

## 4. Discussion

Most proteomics and structural studies on SARS-CoV-2–host interactions have focused on the spike–ACE2 interaction. More data have accumulated around S protein interactions based on its important mechanism. However, with the discovery of other viral proteins’ mechanisms, more interactions with host cell proteins can be investigated. On the enriched cluster, which potentially plays a critical role in viral invasion, the large-scale analysis of variant protein structure docking has not been conducted yet.

In this study, the binding states of SARS-CoV-2 S protein with more known or potential human receptor proteins were achieved. The influence of variant sequences in the docking structures was also considered. The sequence of the Omicron BA.2 variant as a new ligand was implemented in the SARS-CoV-2–human protein docking. A change in the viral proteins’ binding affinity with human proteins was observed.

The SARS-CoV-2–human PPI network is a macroscopic map of the node–node binary associations. It requires combination with the details of protein docking to assess the actual situation of a particular variant protein on viral invasion from a microscopic perspective [53]. The impact of structural changes and the affinity of docking structures through sequence mutations still require further evaluation. With highly accurate methods of molecular dynamics simulations and hotspot predictions [54], more analyses can be conducted on all SARS-CoV-2 proteins and human proteins related to the infection process.

Once the structures of the proteins of emerging SARS-CoV-2 variants are experimentally solved, the association of these proteins with host proteins will be better understood. The topology of the interaction network of each variant strain and the characteristics of the virus-host protein docking interface will provide more help for the prevention and treatment of SARS-CoV-2.

Gene ontology enrichment analysis can also yield enrichment for subcellular locations of human proteins that also need to be analyzed in more detail. Studies of the SARS-CoV-2 interactome [55] have revealed multiple subcellular localizations associated with viral entry, replication, gene expression, and effects on immune responses. Defining the docking events that occur at these subcellular localizations and mining the characteristics of the docking interface under various conditions could help understand the relevant mechanisms of SARS-CoV-2.

In this study, functional analysis and structural docking calculations were performed on the interaction between the viral protein and the host. It was found that the mutations in sequences may influence the structural affinity. The Omicron BA.2 variant has a stronger spike–ACE2 binding affinity as compared to other variants. In binding with AXL and CD209, the BA.1 spike had stronger affinities than the others as well. Compared with BA.2, the S protein of the Delta variant showed weaker affinity with receptors. However, when the Delta variant was compared to BA.1, the opposite relationship was observed in the different protein dockings. Wu et al. [56] provided similar results showing stronger affinity in the Delta S protein RBD binding with human proteins. The results could be related to different single-point mutations in their sequences [57]. The connections of binding affinity with transmissibility, pathogenicity, or stability require further investigation [24].

These results suggested the existence of potential receptors of viral proteins and their binding state alteration with different variants. However, the affinity of the docking structure could also be affected physically or biologically. Combining some specific experimental conditions and more datasets could improve the analysis and is helpful for SARS-CoV-2 protein receptor identification, drug design, and repurposing.

## Figures and Tables

**Figure 1 biomolecules-12-01067-f001:**
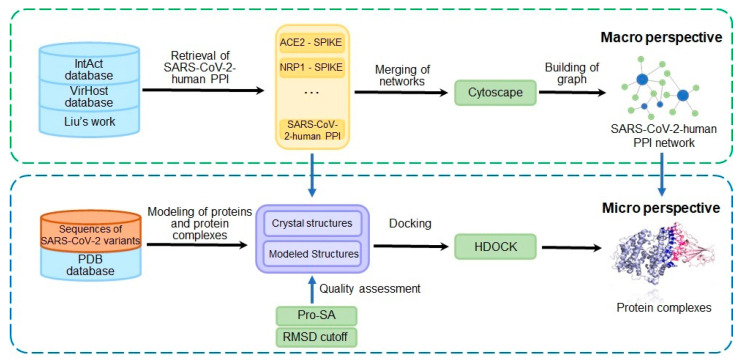
The integration of SARS-CoV-2–human PPI network and virus-host protein structural binding state. The PPI network was merged with experimental PPIs from multiple sources. Binding structures were collected from both experiments and calculations. The 3D view of S protein (Omicron BA.2 variant) and ACE2 protein binding as a demonstration was captured from the structure modeled in this research.

**Figure 2 biomolecules-12-01067-f002:**
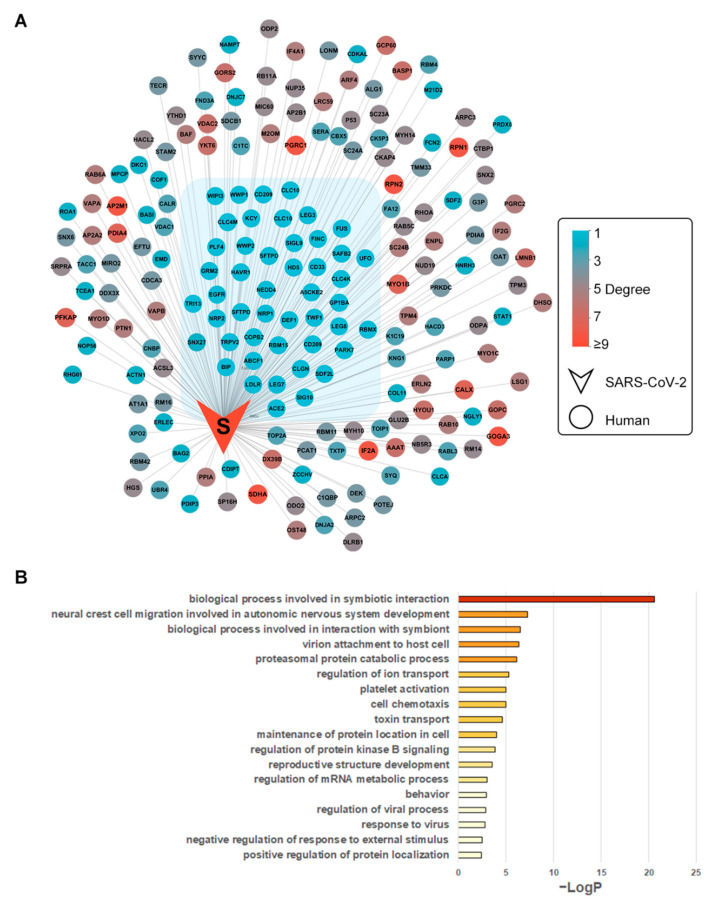
Integrated SARS-CoV-2–human PPI S protein subnetwork and functional enrichment. (**A**) The subnetwork of S protein with its 196 first neighbors. The isolated interactions (degree = 1) were grouped in blue shadow; (**B**) GO enrichment of S protein’s isolated interactions. Nodes represent viral (V-shape) or human (ellipse-shape) proteins. Edges represent protein–protein interactions. The nodes are colored according to their degrees. For human proteins, the degree values range from 1 to 9 (from light blue to red shade). Nodes with lower degree values have fewer physical associations or direct interactions with other SARS-CoV-2 proteins.

**Figure 3 biomolecules-12-01067-f003:**
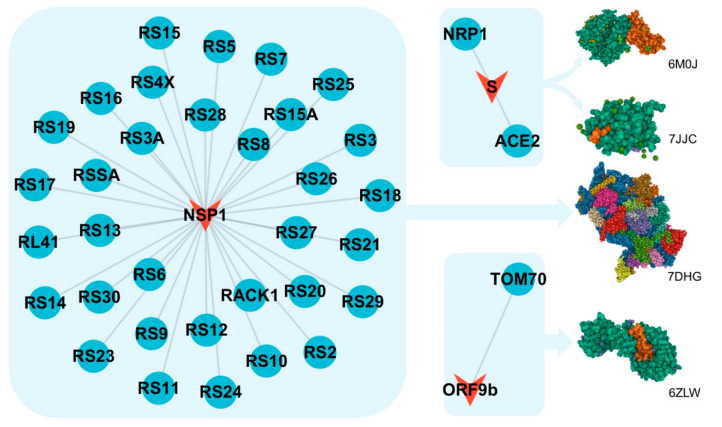
The experimental SARS-CoV-2–human binding structures mapped to PPIs and PPI clusters. Mapped structures are: S protein’s receptor-binding domain (RBD) bound with ACE2 (PDB ID: 6M0J [31]); S protein’s RBD bound with NRP1 (PDB ID: 7JJC [10]); ORF9b complex with human TOM70 (PDB ID: 7DHG [44]); Nsp1 bound to the human 40S ribosomal subunit (PDB ID: 6ZLW [43]). PDB 3D structures were presented by Mol* Viewer [52].

**Figure 4 biomolecules-12-01067-f004:**
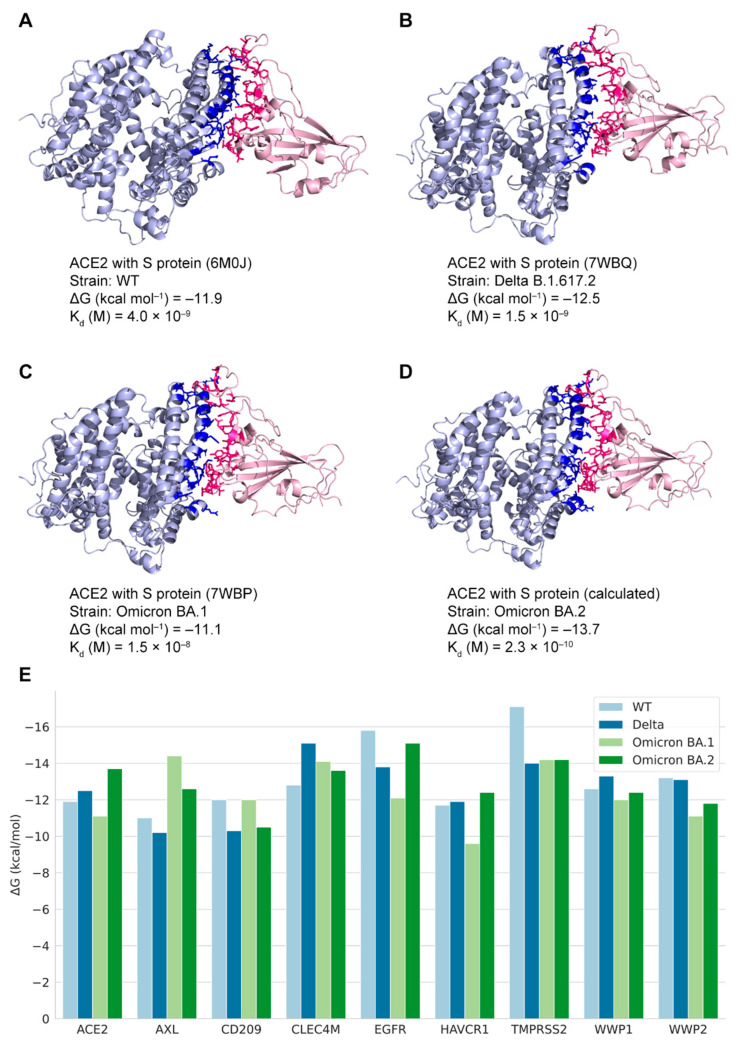
States of human proteins binding with SARS-CoV-2 variants S proteins. (**A**–**C**) Experimental structures of human ACE2 protein binding with S proteins (PDB ID: 6M0J, 7WBQ, and 7WBP); (**D**) Calculated docking structure of human ACE2 protein binding with Omicron BA.2 S protein. The docking temperature is 37 °C. The binding interfaces are highlighted in deep blue and pink colors; (**E**) The ΔG values represent free energy of human proteins binding with variants S proteins.

**Table 1 biomolecules-12-01067-t001:** Structures of SARS-CoV-2 S protein’s isolated interactions.

Gene Name	UniProt	Spike Binding	Template for Docking	Chain	Note
WWP2	O00308		4Y07	A	
NRP1	O14786	7JJC	7JJC	A	
EGFR	P00533		1IVO	A	
AXL	P30530		Modeled structure		Homology template 5VXZ.1.B
HAVCR1	Q96D42		5DZO	A	
ACE2	Q9BYF1	6M0J, 7WBQ, 7WBP	6M0J	A	
WWP1	Q9H0M0		1ND7	A	
CLEC4M	Q9H2X3		1K9J	B	
CD209	Q9NNX6		1SL4	A	
TMPRSS2	O15393		7MEQ	A	

**Table 2 biomolecules-12-01067-t002:** Summary of the quality assessment scores for the models of SARS-CoV-2 S proteins and the interactors.

Model	Percentage in Allowed Region	Z-Score	RMSD	GMQE	Sequence Identity
Spike Omicron BA.2	87.50%	−5.59	0.154	0.75	97.76%
Spike Omicron BA.1	84.10%	−5.81			
Spike Delta	83.90%	−6.04			
Spike WT	89.30%	−5.87			
AXL	95.10%	−4.49	0.168	0.93	95.10%
WWP2	92.30%	−7.68			
ACE2	93.30%	−12.87			
EGFR	66.10%	−8.46			
HAVR1	89.10%	−4.3			
WWP1	85.20%	−8.32			
CLC4M	89.40%	−6.74			
CD209	91.40%	−4.94			
TMPS2	87.90%	−7.11			
NRP1	87.60%	−5.18			

## Data Availability

Data available at http://bis.zju.edu.cn/overcovid/covm2 (accessed on 6 June 2022).

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
