# Peer review of "CoVM2: Molecular Biological Data Integration of SARS-CoV-2 Proteins in a Macro-to-Micro Method"

_biomolecules, 2022, doi:10.3390/biom12081067_

Round 1
Reviewer 1 Report
Dear authors, your article seems reasonable, but it needs to be revised before it gets accepted. My comments are as follows-
1. It is not clear why the authors have provided the name CoVM2
2. The article needs to cite the following references-
Satyam, R., Yousef, M., Qazi, S., Bhat, A. M., & Raza, K. (2021). COVIDium: a COVID-19 resource compendium. Database, 2021.
Barh, D., Tiwari, S., Gomes, L. G., Pinto, C. H. R., Andrade, B. S., Ahmad, S., ... & Uversky, V. N. (2022). SARS-CoV-2 variants show a gradual declining pathogenicity and proinflammatory cytokine spur, an increasing antigenic and antiinflammatory cytokine induction, and rising structural protein instability.
And provide a comprehensive comparison.
3. The sequence similarity in line 92 (template sequence) is missing, and does the authors have checked the model with the Ramachandran plot?
4. The authors need to perform at least 100ns molecular dynamics simulation for all the complexes to check the stability of the complex.
I wish you all the best and want to review the article before its acceptance.
Reviewer 2 Report
The authors of this work developed a protein-protein interaction network that links SARS-CoV-2 related protein interactions. This work also then proposes a method to link structural information to the interactions to get more insight into how the binding interactions are affected y changes in mutations of the SARS-CoV-2 variants. This could be a very valuable technique is expanded computational program and easily accessible to scientists.
Questions
1. How are the protein interactions identified with AP-MS, BioID-MS, and in this work similar? There is such a different number of proteins identified by each of the research groups, can the author provide a perspective on the correlation or implications of these differences and what this means for scientist using this data as a starting point for therapeutic design. What tools can we use to identify the best targets? Below are comments and thoughts that would strengthen the paper.
2. I am a little unclear on the workflow in Figure 1. Can you add annotations in the figure to describe what inputs and outputs are being used and collected from this process? Additionally can you make the words larger in the image?
3. Which is more important strength of interaction of degree of interaction? Can you please speak to this in your paper?
4. Is there any other supporting evidence that can be provided to suggest that the proteins identified in Table 1 would have a role in S protein interactions. Maybe you can look at the SARS-CoV, MERS-CoV, or other coronaviruses to see if there is any background evidence that might support this argument. This would strengthen the paper.
5. The methods needs to be describe more in detail to clearly describe what was used as input source of data, which UniProt sequences, PDB files for the network construction. Was the python script used an in house script made by the authors or a script that can be downloaded. Can a reference be included for the script or the be included in the supplementary data?
6. I think a time stamp should be placed in the methods form line 92. There will be more variant structures solved so this paper should say "at the time of publication" or "at the time of this study was completed" so it is clear years from now, what was not available now might be available then.
7. What parameters where used for SWISS-MODEL to develop the homology models? Please describe what a good metric is for the model in each of the areas listed so it is clear to the readers what criterions where used and should be used in further study.
8. Was the entire spike protein modeled or just the receptor binding domain. In the figures it seems to be only the receptor binding domain. please clarify which part of the SRAS-CoV-2 sequences where included for model development.
9. Why 80% in allowed regions in the Ramachandran plots? This cutoff seems very low. Usually 95% or better is a good cutoff for a homology model.
10. The criterion for a good RMSD is given but not for a good z-score. Please add this to the methods in the paper.
11. In Line 108-110, it states that "Proteins structures that satisfy most or all of the quality assessments will be preserved for downstream docking calculation. Which parameters are sufficient to not be met and still be included in the docking. Which parameters where met for the docking runs in this study? Please classify and justify what can and can not be eliminated.
12. Lines 114-115. For HDOCK scoring the lower the number the better the predicted binding interactions. Can the authors describe why they choose the higher docking scores for further analysis? Please also add details about how HDOCK scores compounds docking poses. In this paper it is 100 models where generated. (Lines 171-172). Which models where used for further analysis?
13. The paper presents only the ACE2 interaction with the receptor binding domain of the Spike protein of various SARS-CoV-2. However there are score for other protein interactions in Figure 4E. I would like to see the interactions to these other less common parings. Please present some of the docking results to these as well, especially since this paper is emphasizing these additional interactions be explored (lines 192-195).
14. I am a little concerned about the discussion. It seems to be a lot of suggestions but not supported by the data in the paper. I would like to see more definitive data form the authors work to support the statements made. especially lines 222-227. No structural data was presented that demonstrated changes the binding regions, thus the authors should present this information to ensure that they can be correlated to the results described.
Comments and changes
1. Line 14. It is not clear what binding interaction is being identified please specify "Binding to what?"
2. Line 18. What do you mean by saying "human" here? Can you please clarify?
3. Line 19. I think this should be "crystal structures" no "experimentally validated docking structures." docking structures are computational only. We can measure binding scores through other biochemical assays.
4. Line 22. You stated that Omicron BA.2 did not have a crystal structure in the manuscript. I might clarify that this structure is a homology model and not a crystal structure that was used.
5. Line 39. I think the second "Omicron BA.1" should be "Omicron BA.2"
6. Line 41. What do "can be obtained from modification" mean? Please clarify
7. Line 47. Please provide a little background on "AP-MS" for readers who will be unfamiliar with this technique. Also please define then abbreviate affinity phase-mass spectrometry (AP-MS).
8. Line 49. Please also describe BioID-MS, and define then abbreviate.
9. Lines 53-54. Please clarify what you mean by "However, these datasets require further integration and adoption of the mutant sequences of SARS-CoV-2 variants?" Do you mean that the variants are not listed in these databases yet?
10. Line 63. I think this should be "crystal structures" instead of "experimentally verified protein docking structures"
11. Lines 63-68 are confusing for me. Can you please rephrase?
12. Line 80. This lines say "mainly" obtained indicating that some data was obtained from different sources. Can you please add the other sources of data for developing the interaction network.
13. Line 117. I think the word applied is not the correct word for this sentence. please rephrase.
14. Line 124-125. What was the reasoning between the SARS-CoV-2 proteins and the human proteins that where included in the network? Please describe the rationale so that other readers can understand the process better.
15. Figure 2. I am not sure what I am supposed to get out of this figure. It is very unclear. There seem to be very small lines of text on the nodes. Can this be enlarged? Is this important to what you are trying to emphasize?
16. Figure 4. Doesn't Omicron BA.1 bind better to the ACE2 than Delta and the WT SARS-CoV-2? If so why does the prediction not align with this? Also the interfaces are so small you can't really glean any information about them? Is there something that we should be looking for, if so can you please create a figure that will make the emphasis clearer?
17. Line 169. Was the entire spike protein used for the docking analysis?
18. I don't quite understand the website interface. It would be helpful for the authors to also provide details about how to use the website as a resource. Please add to the paper details about how to use the website so scientists can use this tool.
Round 2
Reviewer 1 Report
Authors have addressed all my comments and suggestions.
Reviewer 2 Report
Dear Reviewer,
The authors have addressed most of my concerns. Below are the concerns I still have or need further clarification on
1. Can you add to the manuscript in regard to Point 3: Which is more important strength of interaction of degree of interaction? The response you provided below
“The strength of interaction is determined through the detection methods, which can be a measurement of the fact of an exact PPI, or can imply the affinity of the proteins. It’s a valuable parameter but not considered in this manuscript.”
2. I still believe that more clarification is needed behind the template in regard to my original Point 8: Was the entire spike protein modeled or just the receptor binding domain. In the figures it seems to be only the receptor binding domain. please clarify which part of the SRAS-CoV-2 sequences were included for model development. Please add the specific chain number and residue of the chain that was used in which pdb files. This is important because the PDB file for 7WBQ has two a D and B chain which are the RBD but there are a few differences in the orientations of the sides chains which could impact a docking score. For example when these two chains are aligned the side chains of the Q498, N481, K478, YN487, 473, and H519 are slightly different. Please completely describe which chains you used in your modeling process. This is important information to replicate findings.
3. Can you add to the manuscript in regard to Point 9: Why 80% in allowed regions in the Ramachandran plots? This cutoff seems very low. Usually 95% or better is a good cutoff for a homology model a statement to the manuscript about this and then reference it.
4. I do not see that the criterion for a good Z-score is described in the revised version of the manuscript, my original point 10. Please add this statement to the manuscript and then reference it.
5. In regard to my original point 11 “Point 11: In Line 108-110, it states that "Proteins structures that satisfy most or all of the quality assessments will be preserved for downstream docking calculation. Which parameters are sufficient to not be met and still be included in the docking. Which parameters where met for the docking runs in this study? Please classify and justify what can and can not be eliminated” I don’t see that it has been clearly described in the revised version which parameters need to be met to be included for a docking run in this study. Please clearly describe which parameters had to be met.
In regard to my Point 30: “Figure 4. Doesn't Omicron BA.1 bind better to the ACE2 than Delta and the WT SARS-CoV-2? If so why does the prediction not align with this? Also the interfaces are so small you can't really glean any information about them? Is there something that we should be looking for, if so can you please create a figure that will make the emphasis clearer?” I am still a little confused about the delta versus omicron BA.1 can you discuss that further in the manuscript. I only see that the Omicron BA.1 and Omicron BA.2 are described.
